# Individual participant data network meta-analysis of psychosocial interventions for survivors of intimate partner violence: Study protocol

Christina Palantza[1]*, Karen Morgan[1], Nicky J. Welton[1], Hannah M. Micklitz[2‡], Lasse B. Sander[2‡], Gene Feder[1]

1 Department of Population Health Sciences, Bristol Medical School, University of Bristol, Bristol, United Kingdom, 2 Medical Psychology and Medical Sociology, Faculty of Medicine, University of Freiburg, Freiburg, Germany

☯ These authors contributed equally to this work.
‡ These authors also contributed equally to this work.
* christina.palantza@bristol.ac.uk

## Abstract

Many systematic reviews and meta-analyses have been conducted in the field of Intimate Partner Violence (IPV) and the evidence shows small to moderate effect sizes in improving mental health outcomes. However, there is considerable heterogeneity due to variation in participants, interventions and contexts. It is therefore important to establish which participant and intervention characteristics affect the different psychosocial outcomes in different contexts. Individual Participant Network Meta-analysis (IPDNMA) is a gold-standard method to estimate moderating effects, compare the effectiveness of different interventions and thus answer the question of which intervention is best-suited for whom. We will conduct an IPDNMA of randomised controlled trials (RCTs) of psychosocial interventions for IPV survivors aimed at improving mental health, psychosocial outcomes such as self-efficacy and quality of life, reducing IPV and increasing safety-behaviours and dropout from the intervention (as an indication of intervention acceptability) compared to any type of control (PROSPERO registration number: CRD42023488502). We aim to establish collaborations with the authors of eligible RCTs, to obtain and harmonise the Individual Participant Data of the trials. We will conduct one-stage IPDNMA under a Bayesian framework using the multinma package in R, after testing which characteristics of the participants and interventions are effect modifiers. We anticipate that not all study authors will provide access to IPD, which is a limitation of IPDNMA. We aim to address this by combining studies with aggregate data and studies with IPD using Multi-Level Network Meta-Regression (ML-NMR) implemented in the multinma R package. This approach is novel in the field and makes full use of available evidence to inform clinical and policy-related decision making.

**Data availability statement:** Data availability is not applicable to this article as it is a protocol merely outlining research that has not been conducted yet. Upon completion of the study, it will still not be possible to make the data publicly available, as it is highly sensitive and it belongs to the researchers that conducted the eligible studies; they are to share them with us only for the purposes of this study.

**Funding:** Christina Palantza is in receipt of an annual postgraduate research stipend from the University of Bristol from 2023 to 2027. There is no specific grant number. This is the only funding source of this review. The funder had no role in the development of this protocol. The funders had no role in study design, data collection and analysis, decision to publish, or preparation of the manuscript.

**Competing interests:** I have read the journal's policy and the authors of this manuscript have the following competing interests: Christina Palantza is in receipt of an annual stipend from the University of Bristol from 2023 to 2027. This does not alter our adherence to PLOS ONE policies on sharing data and materials. CP's scholarship granted by the University of Bristol does not constitute a competing interest. Role of funder: The funders had no role in study design, data collection and analysis, decision to publish, or preparation of the manuscript. Non-Financial competing interest: None.

## Introduction

The term Intimate Partner Violence (IPV) covers all forms of aggression towards a current or former intimate partner, including psychological, physical, sexual, or coercive controlling behaviours [1]. One of the most severely impacted aspects is the mental health of the survivors [2]; in fact mental disorders are more common among both men and women victims/survivors and perpetrators of IPV than among the general population [3]. The course of IPV affects the course of mental distress and vice versa [4]. IPV is a worldwide public health issue [5], and it has highly adverse outcomes for victims/survivors, perpetrators and children [6,7,8]. IPV was further exacerbated during the COVID-19 pandemic [9]. This increase has been consistently associated with a generally well-established risk factor for IPV, which is financial strain [10,11]. Therefore, it is likely to continue rising during the cost of living crisis starting in early 2022, which is affecting many regions around the world [12], and non-governmental organisations working with survivors are already reporting the negative impact of the crisis [13]. There is thus a heightened necessity for efficient services that are targeted to meet individuals' needs by utilising only the resources that are needed.

Despite the heightened need, psychosocial care for survivors appears inadequate both in terms of accessibility and quality [14]. The psychosocial interventions tested in research settings show some potential [15], but effect sizes are rather small (i.e., Standardised Mean Difference of −0.15 to −0.26) for major psychological outcomes such as depression, or inconsistent across reviews for re-victimisation, and both clinical and statistical heterogeneity of the intervention effects are high [15,16]. This could be attributed to heterogeneous populations, both in terms of factors related to the context of the relationship (i.e., dating, or parenting children together) and belonging to minoritised demographic groups [17]. Moreover, the quality of the evidence is questionable, as many studies do not use the most methodologically robust design of a randomised and controlled trial [18], and those that do have not reported all the descriptive statistics that are necessary for a meta-analysis for all the outcomes [16]. In addition, many important aspects, such as long-term effects, adverse effects of the interventions, e.g. increasing mental distress or IPV victimisation, and differential effects for survivors with different characteristics from different contexts remain unanswered, despite the indications towards considerable impact of intersectionalities [16,18]. This is particularly problematic, because the survivors are indeed quite diverse, and the experiences of minoritised groups that face discrimination are different [19], and come from different contexts (e.g., staying with the perpetrator, staying in a shelter, co-parenting with the perpetrator) indicating different needs from psychosocial support [19,20].

Micklitz and colleagues in their meta-analysis, which is one of the most recent and comprehensive meta-analyses on IPV and mental health, categorised the tested psychosocial interventions into three main categories: psychological, advocacy-oriented and integrated (a combination of the two) [16]. Psychological interventions are psychotherapeutic interventions that employ cognitive, behavioural or emotion-focused techniques, and are usually delivered by a therapist through talking. More specific categories of psychological interventions are cognitive-behavioural, third wave of cognitive behavioural therapy, systemic, and others, such as psychodynamic. Advocacy interventions could be summarised as advice on practical matters and continued support and counselling, enhanced access to other resources such as shelters, and facilitated safety planning [17]. A realist review of advocacy interventions highlighted the critical impact of the nuances both among survivors and contexts, and the importance of matching the intervention to the survivor's context and needs [17]. However, since this study was a realist review, it was not appropriate to examine heterogeneity quantitatively. The heterogeneity of interventions has not allowed conclusions on the effectiveness to be

drawn for large groups of survivors, such as mothers [21], and also leaves the question of what is the effectiveness of each set of intervention components unanswered or might even lead to erroneous conclusions of lack of effect in conventional meta-analyses that lump all interventions together [22].

One of the main sources of heterogeneity in interventions is the different components they include, even when they belong to the same type of interventions, because both advocacy interventions and psychotherapy can be characterised as complex interventions. Complex interventions include multiple, often interacting, components [23]. There is some evidence pointing to possible interactions between the components of psychosocial interventions for IPV interact, more specifically interventions integrating advocacy and psychotherapy were found to be most effective in the meta-analysis by Micklitz et al [16]. There are indications that commonly tested interventions, such as advocacy and cognitive behavioural therapy reduce physical and psychological IPV, but do not reduce sexual IPV and overall IPV [24], and these effects should be investigated in further detail, as heterogeneity is an important limitation for these outcomes too. This might also explain the contradictory evidence on effectiveness of web-based interventions [25,26].

A methodological approach that is well-suited for investigating such nuances in terms of population and intervention, is Individual Participant Data meta-analysis (IPDMA). This form of meta-analysis pools individual level data instead of the aggregate data from the existing studies, by requesting the datasets with the raw data from each eligible study and harmonising the variables, and statistically combining them whilst respecting the randomisation in each study [27]. Meta-analyses of aggregate data are limited to the outcomes reported in publications and can only explore study-level moderators which often lack statistical power and are vulnerable to ecological bias, where covariate effects at the study-level do not accord with those seen at the individual level [28]. IPDMA overcomes these limitations by allowing outcomes and effect modifiers to be comparable across studies, enabling testing and adjusting for various within and between study moderators, both participant and intervention-related, to establish the effect sizes for different interventions and survivors [28–30]. Importantly IPDMA has increased statistical power compared to the original studies by pooling effects. This way, the findings of that participant and counselling characteristics have differential impact [31] can be further explored.

Given the limited resources of health systems [32] and the different interventions tested for IPV [16], it is promising to compare the effectiveness of different interventions, and identify the best-suited intervention for an individual based on their characteristics and context. A method that compares multiple different interventions and ranks their effectiveness is Network Meta-analysis (NMA) [33]. This is achieved by forming a network of all the interventions that have been tested for an outcome and combining direct comparisons of interventions (i.e., compared in the same trial) with indirect ones (i.e., two interventions have not been tested in the same trial, but they were tested against the same control condition), thus allowing for estimation of relative effects of pairs of interventions and hierarchical ranking [34]. The key assumption is that included studies do not differ in terms of factors that could impact the relative effects and have not been adjusted for in the analysis [34]. If this assumption does not hold then there can be inconsistency, which occurs when the direct and indirect evidence give contradictory results. If IPD are available, then effect modifying effects can be estimated and adjusted for to increase validity of the estimates and also increase precision [35]. Individual Participant Data Network Meta-analysis (IPDNMA), which combines IPDMA and NMA to explore individual and study-level moderators for multiple interventions, and can provide estimates of intervention effectiveness for different populations [36]. In addition, this approach also enables the estimation of different components of interventions [23].

IPDNMA has the potential to address many of the limitations and issues raised by one of the most recent meta-analyses with a very close scope to this one [16]. Micklitz and colleagues found that psychosocial interventions, and especially interventions combining advocacy with psychotherapy, i.e., integrative interventions, had a small yet statistically significant effect in reducing depression and post-traumatic stress symptoms, compared to waiting list or inactive control conditions. However, there was considerable heterogeneity even in the subgroup analyses, and half the studies could not be included in the quantitative meta-analysis because the necessary statistics were not reported at the publications of the primary studies [16]. In addition, the authors highlighted the need to test participant-level moderators of effect [16]. We aim to build on the comprehensive scope and rigour of the study by Micklitz and colleagues by updating and adapting their searches, and seeking IPD from the included studies. Given that most RCTs have used Treatment As Usual (TAU) control conditions, or participants were most commonly allowed to seek other help, and also that we are going to compare interventions, we are going to estimate the effectiveness of the interventions, i.e. compared to a real-world standard, rather than efficacy, i.e. that the interventions have some effect compared to nothing, which is usually a sterilised control condition.

## Aim and research questions

This study aims to provide more valid and detailed estimates of the comparative clinically relevant benefits and negative effects of each type of psychosocial interventions than the existing pairwise meta-analyses. The IPD of the studies will be obtained to enable this estimation, as well as exploration and adjustment of effect modifiers. We will fulfil this aim by conducting an IPDNMA, to estimate the relative effectiveness of multiple psychosocial interventions for mental health, psychosocial well-being, IPV, safety behaviours, intervention dropout and harms (i.e., adverse effects of the interventions). We also aim to estimate study-level and individual-level moderator effects to identify which psychosocial intervention suits individual IPV survivors best. More specifically the following research questions will be answered:

1. Which victim/survivor and intervention characteristics moderate the comparative effectiveness of each set of intervention components that have been tested together in improving commonly tested mental health and psychosocial well-being outcomes of IPV victims/survivors?

2. Which victim/survivor and intervention characteristics moderate the comparative effectiveness of each set of intervention components that have been tested together in reducing IPV and improving safety behaviours of IPV victims/survivors?

3. Which victim/survivor and intervention characteristics moderate the comparative effectiveness of each set of intervention components that have been tested together in improving service use of IPV victims/survivors?

4. Which victim/survivor and intervention characteristics moderate the dropout from each intervention?

## Materials and methods

### Design

The study design is an IPDNMA, updating and extending the systematic review of Micklitz and colleagues. IPDNMA makes less strong assumptions than other methods, but does require more data to be able to fit it. We therefore also conduct a conventional meta-analysis,

an aggregate data network meta-analysis, and an individual participant data meta-analysis, to explore the impact of assumptions in these methods and the robustness of the findings to methodological approach. A Patient and Public Involvement group of survivors of intimate partner violence from the UK was consulted and provided input on conceptual aspects.

## Protocol registration

The study protocol has been preregistered on PROSPERO (CRD42023488502) and a project has been created on the Open Science Framework https://osf.io/72uwe/?view_only=1ba29 0a378514c3a929d7eac035bfd67. The Preferred Reporting Items for systematic reviews and meta-analyses (PRISMA) statements for systematic review protocols (PRISMA-P) [37] are followed (see appendix).

## Eligibility criteria

The eligibility criteria based on the PICO (Participants, Interventions, Comparators and Outcomes) are listed in Table 1.

Only randomised controlled trials (RCTs) are eligible for inclusion, including pilot RCTs and studies with few participants, as in most cases they will be analysed together with larger studies testing a similar intervention. If however a smaller study has tested a very different psychosocial intervention, it will be formally included, and it will be investigated if its components match any of the other components, otherwise it will not be possible to include it in the quantitative synthesis. All papers published in peer-reviewed journals are eligible, as well as protocols and trial registrations in case the authors already have data that they can provide. No language or publication date restrictions are applied.

## Identification of studies-information sources

A search strategy with keywords and phrases such as "intimate partner violence" and "psychotherapy" will be used to search the Web of Science database from inception to August the 30th 2024, and PsycINFO, Medline, Embase, and Cochrane Central from March the 23rd 2022 to August the 30th 2024 to update the search of Micklitz and colleagues. The full search string is presented in Table 2.

This search strategy was used by Micklitz and colleagues, so an update of their search will be performed in the databases they searched (PsycINFO, Medline, Embase, and Cochrane

**Table 1. Eligibility criteria by PICO.**

| | |
|---|---|
| **Participants** | Individuals or couples reporting either past or ongoing IPV at randomisation, but studies solely on those involved in perpetration will be excluded. Studies will be deemed partially eligible if they did not recruit participants based on IPV status, but they measured IPV at baseline, and a proportion of the participants had been victimised at baseline. The IPD only of the victimised participants will be requested. If it is not possible to obtain the IPD of a partially eligible study, and outcome measures separately for victimised participants are not reported in the publication, the study will not be included in the analysis at all. |
| **Interventions** | Psychosocial interventions aimed at IPV survivors using any mode of delivery at any setting. Both psychotherapeutic and advocacy interventions will be deemed as psychosocial. Interventions aimed solely at perpetrators, primary prevention to individuals who have not experienced IPV at baseline, and physical safety (such as being in a refuge/shelter and police interventions) will not be included. |
| **Comparators** | All comparators are eligible, including no intervention, waiting list, Treatment as Usual (TAU), another active intervention, placebo or sham version of the intervention. The interventions will be categorised and classified into nodes of the network based on their description regardless of whether they were tested as control or active condition in the original studies. |
| **Outcomes** | Eligible outcomes are all outcomes related to mental health (e.g., depression) as well as psychosocial outcomes (e.g. empowerment and social support). Studies measuring only IPV are not eligible, but if the included studies have measured IPV, it will be analysed. Likewise, studies are not included based on whether they measured intervention dropout, and health service use, but these outcomes will be analysed. |

Central), and additionally on Web of Science. Clinical trial registries will be searched, and forward and backward search will be performed (i.e. the reference lists of the included studies will be checked for eligible papers, as well as which studies cite the eligible studies).

## Study selection process

Two independent researchers will screen the titles and abstracts. All completed studies included in the systematic review of Micklitz et al will be included. The authors of all the protocols and trial registrations identified by Micklitz and colleagues will be contacted to find out whether data are available, and if they agree to share their data, they will be offered co-authorship of the publications derived from the study. If the studies or protocols included at this stage have been published, the full text will be screened by the two independent researchers. In the case of trial registrations, the complete registrations will be screened by the two independent researchers, or a detailed description will be requested from the principal investigators of the studies. In case of disagreement between the two researchers, a third, senior member of the team will be consulted. The screening and inclusion of abstracts and full texts will be done using Rayyan [38].

## Ethical considerations

Since this type of study is essentially a secondary analysis of existing data, it does not need to be ethically approved. However, the trialists of the eligible RCTs are responsible for the ethical re-use of the data they collected, so it is up to them to either consult their local Ethical Review Boards and possibly to submit this re-use for ethical approval, or to request us to sign a Data Sharing Agreement with them. The secure handling of this sensitive type of data is described in the following section "Data Management".

## Data management

A collaborative project will be created on the Research Data Storage Facility (RDSF) of the University of Bristol, managed by GF and accessible to the rest of the authors. The data can be uploaded to that space by the authors of the primary studies. This cloud is compliant with the General Data Protection Regulation. In case the authors face unresolvable issues with this cloud, secure sharing links to a private folder at the institutional OneDrive of CP will be generated. As soon as the data have been uploaded, they will be saved directly in the RDSF and removed from OneDrive. The data will be accessed through the university-managed and password protected device of CP for analysis. Since the data are sensitive, a Data Protection Impact Assessment will be performed.

**Table 2. Full search strategy.**

intimate partner violence OR spouse abuse OR domestic violence OR gender-based violence OR rape OR battered women OR "intimate partner violence" OR "partner violence" OR "interpersonal violence" OR "dating violence" OR "domestic violence" OR "family violence" OR "gender-based violence" OR "gender based violence" OR "violence in close relationships" OR "intimate partner abuse" OR spous* abuse OR "abusive relationship" OR "marital abuse" OR "marital rape" OR "dating abuse" OR "intimate partner aggression" OR "battered women" OR "battered wives" AND Mental health services OR Psychotherapy OR Couples Therapy OR Counselling OR Psychosocial intervention OR Social Work, Psychiatric OR Psychiatric Rehabilitation OR Therapy, computer-assisted OR Telemedicine OR Internet-Based Intervention OR Distance counselling OR "intervention" OR "treatment" OR "program" OR "therap*" OR "psychotherap*" OR "counsel*" OR "support service*" OR "advocacy" OR "secondary prevention" OR "tertiary prevention" OR "rehabilitation" AND randomized controlled trials as topic OR clinical trials as topic OR "randomized controlled trial" OR "clinical trial" OR "controlled clinical trial" OR "clinical trial" OR "clinical trial protocol" OR "clinical study" OR RCT OR random* OR trial

## Data extraction of study characteristics and aggregate data

The study characteristics to be extracted in a spreadsheet are: author, publication year, citation, country, recruitment setting, trial setting, number of trial arms, number of follow-ups, timing of follow-ups, timing of post-treatment, primary and secondary outcomes, type of randomisation, unit of allocation, method of randomisation, sample size in total and in each arm, treatment and study dropout, intention-to-treat or per-protocol analysis, age, gender, current frequency and/or severity of IPV, past frequency and/or severity of IPV, specific target population, eligibility criteria, description of intervention, theoretical orientation, delivery mode, type of therapists/facilitators, number and duration of sessions, frequency of sessions, description of control, whether adverse events and adherence to the intervention were reported, conclusions, the means and standard deviations of all relevant outcomes at all timepoints, or which statistics were reported instead of mean and standard deviation. The characteristics and aggregate data that have already been extracted by Micklitz et al [16] for the studies they included in their meta-analysis will be used in our analysis.

## Data collection

The corresponding authors of eligible studies will be emailed and invited to share the individual-level data of their study. The most recent email address of the author will be sought. If necessary, a reminder will be sent after three weeks and six weeks. If unreachable, the same process will be followed for the senior author of the study. If no valid email address of neither the corresponding nor the senior author can be found, social networking websites such as ResearchGate and LinkedIn will be used to reach the authors, and the same number of reminders will be sent if necessary. The IPD will be deemed unavailable if no response is received after three weeks of the second reminder to the senior author (12 weeks in total). After authors have accepted to share the IPD, there is no specific time limit to send the data.

## Data items

It is expected that not all eligible studies have collected data on all possible moderating characteristics, such as income level, and in such a multifaceted issue, there can be various potentially impactful variables. Therefore, the complete datasets will be requested including baseline and all available follow-ups, with an exception for variables that could lead to the identification of a participant and violate anonymity.

## Data harmonisation

A standard coding of the characteristics of participants will be established based on the operationalisation used by the majority of the studies and the data of each individual study will be recoded accordingly. We currently propose the operationalisation below, but the final one will depend on the data. If a study has not collected any data on some of the listed demographic or socio-economic characteristics, this will be taken into account in the assessment of certainty of the evidence.

1. Biological sex: female/male/intersex

2. Gender identity: cisgender/transgender/gender non-conforming

3. Sexual orientation: heterosexual/homosexual/bisexual/other

4. Age at baseline: continuous variable

5. Relationship status: being in a relationship/not being in a relationship/not sure

6. Cohabitation status: cohabiting with partner/non-cohabiting with partner

7. Parenthood: parent living with children/parent having some contact with their children/ parent having no contact with their children/non-parent

8. Education: Uneducated/ Primary education/ Secondary education/ Tertiary education/ Other

9. Employment status: student/employed/unemployed seeking employment/unemployed not seeking/retired or disability

10. Income level: OECD (Organisation for Economic Co-operation and Development) quintiles

11. Ethnicity: categorical variable with the respective ethnicities in the respective country/trial

12. Presence of psychiatric/developmental disorder

13. Presence of history of abuse in the family

14. Presence of substance dependence

The data from different validated scales will be divided by the standard deviation. If a study uses more than one scale for the same outcome, these scales will be combined in a "within-study-synthesis", by using meta-analytical methods [39]. If any scale on characteristics of the therapists, such as level of empathy, is available, it will be used.

## IPD integrity

In order to check the integrity of the data received, a replication of the summary statistics and main analysis of each study will be attempted by two researchers, and if it fails, the authors will be asked for clarification. In addition, conventional MA with the aggregate data will be performed to replicate the existing findings, and the statistical package that we will use (multinma) is able to integrate the studies that provide IPD with those that do not [40]. More details are provided in the statistical analysis section.

## Risk of bias and assessment of certainty of the evidence

The risk of bias of the studies will be assessed using the Cochrane Risk of Bias Assessment 2.0 [41], which assesses bias arising from the randomization process, bias due to deviations from the intended interventions, bias due to missingness of outcome data, bias in the measurement of outcome, and bias due to selective reporting of outcomes. The overall certainty of the evidence will be assessed with the Grading of Recommendations Assessment, Development and Evaluation (GRADE)—Guidelines [42], which, apart from the risk of bias, assesses inconsistency, indirectness, imprecision and overall quality of the evidence. These assessments will be performed by two independent researchers and disagreements will be resolved through discussion with a third more senior researcher. For the studies already assessed by Micklitz and colleagues [16], the domains of bias arising from missing data and selective outcome reporting will be revisited in case the IPD can elucidate previously unclear points. Publication bias will be assessed through a funnel plot for each outcome, but asymmetry will not be interpreted as publication bias, as this is not appropriate for network meta-analysis [43]; instead appropriate network meta-regression will be used.

## Categorisation of interventions and control conditions

Based on the work of Micklitz and colleagues, the interventions that have been tested so far can be categorised into the following nodes:

1. Advocacy interventions: interventions based on the empowerment theory of Dutton [44] and the stages of change model [45], where facilitators use non-directive techniques such as motivational interviewing to guide the survivors to reach specialised services (also legal and financial ones), and to establish a safety plan

2. Integrative interventions: interventions combining the elements of the advocacy interventions with those of the psychological ones, including safety planning

3. Cognitive behavioural interventions: psychological interventions employing cognitive and behavioural techniques, such as cognitive restructuring

4. Third wave therapies: psychological interventions with more targeted cognitive approaches, such as mindfulness

5. Systemic interventions: family or couple based psychological interventions

6. Other psychological interventions: other psychological approaches tested by a small number of studies, such as interpersonal, humanistic or meditation based

The control conditions are expected to be categorised for the NMA as follows, unless different clusters of control conditions with common components appear in the final set of eligible studies:

1. Waiting list

2. No intervention

3. Information on services

4. Referral to services

5. Active psychiatric care

6. Contact

7. Community care

8. Support groups

9. Hotlines

However, small changes might be made depending on the components included in the control conditions of the included studies. Additionally, we will also compare different components of interventions, in order to provide further insight into the exact active mechanisms that lead to the effects and potentially interact with participant characteristics. We will define these components through a process of qualitative mapping, and we will attempt to involve the trialists of eligible studies in this process, as they are likely to be experts in these interventions.

## Outcomes and effect measures

There is a vast array of mental health, psychosocial, IPV and safety-behaviour outcomes of interest tested by the primary studies, which introduces heterogeneity in the estimation of effectiveness of interventions, and thus limits certainty, hampers interpreting the estimates accurately and establishing clinical relevance, but it is the reality of the field [46]. Psychosocial interventions use primarily psychological techniques that target mental health symptoms [16]. So, in our study, the primary outcomes are the most commonly measured mental health outcomes, as the results of the Micklitz et al meta-analysis showed that there is not sufficient

evidence on other outcomes to allow for evidence synthesis. These commonly measured outcomes are symptoms of:

1. depression

2. anxiety

3. PTSD

4. Substance use

The secondary psychosocial outcomes are: suicidal ideation, perceived self-efficacy, self-esteem, quality of life, social support, decisional conflict, empowerment, all as scores in validated self- or clinician-rated scales, clinical interviews, or composites of scales. The main timepoint is post-treatment, but all available follow-ups will be extracted, and if there is sufficient homogeneity in the timing of follow-ups, analyses will be conducted.

Additionally, the IPV and safety behaviour outcomes are: continuation of frequency/intensity of IPV, as a whole (using the total scores of scales), and also the subtypes of psychological, physical, sexual abuse and coercive control separately when available (as scores on subscales), which will all be operationalised continuously as scores on validated (sub)scales. Safety-related behaviours will be examined only if there is not too much heterogeneity in the definition and operationalisation of this outcome by the primary studies.

It is also interesting to investigate the effects of the interventions on service use, and acceptability of the interventions. Service use will be operationalised as scores on checklists. Satisfaction with the treatment will also be operationalised as scores on respective questionnaire items. Acceptability of the interventions will be defined as the number of participants dropping out of the intervention (not necessarily the whole study and assessments) (dichotomous outcome). The effect size for all continuous outcomes will be Hedge's g, and for all dichotomous outcomes the odds ratios will be calculated. Like for primary outcomes, the main timepoint of interest is post-treatment, but all available follow-ups will be extracted, and if there is sufficient homogeneity in the timing of follow-ups, analyses will be conducted.

## Statistical analysis-synthesis

The analysis will be done gradually from the simplest to the most complicated model. Firstly, a conventional meta-analysis excluding studies comparing active psychosocial interventions will be conducted to establish the overall effectiveness of psychosocial interventions. Secondly, an aggregate data network meta-analysis will be conducted to compare and rank the effectiveness of each psychosocial intervention; the active interventions that were tested as controls in the original studies will be classified in the node of active interventions they share the same components and approaches with. Thirdly, a component network meta-analysis will be conducted to compare and rank the effectiveness of the different components or sets of components that were tested together. Fourthly, an Individual Participant Data Meta-analysis will be conducted (excluding the studies that compare two active psychosocial interventions) to estimate moderator effects on relative effectiveness of psychosocial interventions as a whole. Given that numerous potential moderators are likely to be available, we will explore if it is possible to reduce them by running collinearity diagnostics. Missing data will be imputed for each study using multiple imputation. We will also create pseudo-IPD by simulating from a model using the reported baseline summaries and regression coefficients for effect modifiers as described in [47], if available in the publications of studies that do not provide IPD.

The same procedure will be repeated including all studies for the Individual Participant Data Network meta-analysis. We will assess whether survivor and setting characteristics (such

as being separated or still living together with their partner, and clinical or community settings) are effect modifiers or prognostic factors, based on results of the IPDMA (i.e. whether there are interactions between intervention and characteristics) and the following stages of the analysis will be divided into subgroups of studies that are similar in effect modifiers, so that the transitivity assumption is met. An one-stage IPDNMA will be conducted under a Bayesian framework using the multnma package in R. The characteristics of the interventions, such as number of sessions, format and mode of delivery will be tested as effect modifiers.

We will also conduct an analysis to include studies that do not provide IPD or sufficient information to create pseudo-IPD using multilevel network meta-regression (ML-NMR implemented using multinma in R) [40]. ML-NMR is a method that can combine studies with IPD and studies with aggregate data summaries by defining a participant-level regression. Studies with IPD directly inform the individual-level model, whereas studies with aggregate data inform an integral over the model based on the covariate distribution of each study [40].

In all of the aforementioned models where this is possible (aggregate data NMA, IPDMA, IPDNMA), if there is sufficient evidence we will explore fitting component (IPD) NMA models [23,48,49]. A sensitivity analysis excluding studies with high risk of bias will be conducted separately for each bias domain. The rankings of the interventions and the Surface Under the Cumulative Ranking (SUCRA) scores will be calculated to summarise the results of the IPDNMA.

For the outcome of intervention dropout we will synthesise absolute treatment effects, as many control conditions are inactive, by modelling the probability of dropout through a generalised linear model with a binomial likelihood, as in [40].

Heterogeneity will be assessed by comparing the fit of fixed and random effects models, reporting between-study variance, $\tau^2$, and explored through inspection of regression coefficients and sub-group analyses. Inconsistency will be assessed by fitting a model that relaxes the consistency assumption, and if necessary explored further using a node-splitting model [33].

Given the sheer number of intervention and moderator effects that we will estimate, we acknowledge that the probability of chance findings increases. Therefore the results will be interpreted appropriately conservatively.

## Discussion

We plan to conduct an IPDNMA to estimate precisely the effects of psychosocial interventions on the mental health of IPV survivors, to compare the interventions and their components, and to examine potential moderating effects of the characteristics of the participants and the interventions. We intend to fulfil this aim by establishing a collaboration with the authors of the RCTs. This work will disentangle a considerable part of the heterogeneity in the field of psychosocial interventions for IPV victims/survivors. More specifically, the moderating effects of key participant characteristics will be investigated, which thus enhances the evidence on whether intervention effectiveness differs for survivors with different and perhaps intersecting identities. Moreover, we will examine more thoroughly which components of the interventions precisely are of particular importance.

### Strengths and limitations

This methodology is considered a gold standard in evidence synthesis [29] and it is novel in the field of IPV. The synthesis approach allows research findings to be integrated to increase precision [28], whilst thoroughly investigating factors affecting the effects of the interventions on the outcomes with advanced statistical methods. These factors will pertain to individuals,

interventions and contexts, and possibly their interaction, to address heterogeneity as fully as possible. The certainty of the evidence will also be assessed under the GRADE method [42].

However, there are admittedly limitations to our approach. We will be limited by the data measured in the included studies, and statistical power is bound to vary across studies and outcomes. In addition, it may not be possible to harmonise outcomes across studies that have measured them in different ways, which means that some facets of the concepts included only in measures rarely used might be missed. We plan to address as much as possible by sensitivity analyses. The different evidence synthesis methods make weaker or stronger assumptions, but these come down to there being no differences in effect modifiers between the included studies that have not been adjusted for. The more flexible models using IPD are more able to adjust for effect modifiers more effectively, and so make less strong assumptions. The IPD-NMA method allows us to assess the main assumptions and also gives flexibility in the extent that the assumptions are applied [40]. The most common issue of IPDMAs is that the IPD are often impossible to obtain [50]. This can be a serious caveat depending on the proportion of unavailable studies, as lack of IPD limits the flexibility of IPDMAs. We plan to deal with this issue by attempting to contact the researchers through a variety of different channels (not just via email), issuing project newsletters, and holding collaborator meetings to encourage engagement. For studies where we do not have IPD we will endeavour to still include them by using pseudo-IPD if possible [47], and if not, by using multi-level network meta-regression, which enables synthesis of evidence from studies with IPD and aggregate data, overcomes aggregation bias and non-collapsibility bias [40].

Efforts will be made to reduce publication bias by expanding the literature search of Micklitz and colleagues to the Web of Science database, and searching trial registries and grey literature.

## Relevance

The field of IPV is highly heterogeneous, both because IPV affects diverse individuals and because the existing research has tested various interventions and outcomes with different measures, so matching individuals and contexts with interventions is crucial [17]. This has not been done yet with quantitative methods that limit personal biases and offer precise estimates of the impact of each of the many factors intertwined in the provision of psychosocial support of IPV survivors. Pinpointing impactful factors and comparing the existing interventions can contribute to more efficient decision making both for practice and policy.

**Changes to the protocol:** Since the awarding of the funding and the beginning of the study, a few changes have been made to the protocol. The data reduction method has changed from a lasso method to assessment of collinearity, and more emphasis has been placed on the comparison of intervention components, as it appears to be important for the overall objective of disentangling effectiveness.

## Supporting information

**S1 Checklist. PRISMA-P checklist.**
(DOCX)

## Author contributions

**Conceptualization:** Christina Palantza, Hannah M. Micklitz.

**Data curation:** Hannah M. Micklitz, Lasse B. Sander.

**Funding acquisition:** Gene Feder.

**Methodology:** Nicky J. Welton.

**Project administration:** Christina Palantza.

**Resources:** Hannah M. Micklitz, Lasse B. Sander.

**Supervision:** Karen Morgan, Nicky J. Welton, Gene Feder.

**Writing – original draft:** Christina Palantza.

**Writing – review & editing:** Karen Morgan, Nicky J. Welton, Hannah M. Micklitz, Lasse B. Sander, Gene Feder.

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
