## [Decision Letter · Decision Letter 0]

30 Oct 2024

PONE-D-24-25327Individual Patient Data Network Meta-analysis of psychosocial interventions for survivors of intimate partner violence: Study protocolPLOS ONE

Dear Dr. Palantza,

Thank you for submitting your manuscript to PLOS ONE. After careful consideration, we feel that it has merit but does not fully meet PLOS ONE’s publication criteria as it currently stands. Therefore, we invite you to submit a revised version of the manuscript that addresses the points raised during the review process.

 The referees have provided detailed comments that are extremely helpful, and highlight that more specificity as well as some textual editing is needed in various parts of the manuscript.  Given the high quality of the referee reports I will not add further comments but request that you respond carefully to these two reports.

A marked-up copy of your manuscript that highlights changes made to the original version. You should upload this as a separate file labeled 'Revised Manuscript with Track Changes'.An unmarked version of your revised paper without tracked changes. You should upload this as a separate file labeled 'Manuscript'.

We look forward to receiving your revised manuscript.

Kind regards,

Jessica Leight, PhD

Academic Editor

PLOS ONE

Journal requirements:  When submitting your revision, we need you to address these additional requirements. 1. Please ensure that your manuscript meets PLOS ONE's style requirements, including those for file naming. The PLOS ONE style templates can be found at  https://journals.plos.org/plosone/s/file?id=wjVg/PLOSOne_formatting_sample_main_body.pdf and  https://journals.plos.org/plosone/s/file?id=ba62/PLOSOne_formatting_sample_title_authors_affiliations.pdf  2. Thank you for stating the following in the Competing Interests section:  [I have read the journal's policy and the authors of this manuscript have the following competing interests: Christina Palantza is in receipt of an annual stipend from the University of Bristol from 2023 to 2027.].  Please confirm that this does not alter your adherence to all PLOS ONE policies on sharing data and materials, by including the following statement: ""This does not alter our adherence to  PLOS ONE policies on sharing data and materials.” (as detailed online in our guide for authors http://journals.plos.org/plosone/s/competing-interests).  If there are restrictions on sharing of data and/or materials, please state these. Please note that we cannot proceed with consideration of your article until this information has been declared.  Please include your updated Competing Interests statement in your cover letter; we will change the online submission form on your behalf. 

 3. Thank you for stating the following financial disclosure:   [Christina Palantza is in receipt of an annual postgraduate research stipend from the University of Bristol from 2023 to 2027. There is no specific grant number. This is the only funding source of this review. The funder had no role in the development of this protocol.].   Please state what role the funders took in the study.  If the funders had no role, please state: ""The funders had no role in study design, data collection and analysis, decision to publish, or preparation of the manuscript.""  If this statement is not correct you must amend it as needed.  Please include this amended Role of Funder statement in your cover letter; we will change the online submission form on your behalf. 4. Please amend your manuscript to include your abstract after the title page.

Reviewers' comments:

Reviewer's Responses to Questions

**Comments to the Author**

1. Does the manuscript provide a valid rationale for the proposed study, with clearly identified and justified research questions?

Reviewer #1: Yes

Reviewer #2: Yes

2. Is the protocol technically sound and planned in a manner that will lead to a meaningful outcome and allow testing the stated hypotheses?

Reviewer #1: Yes

Reviewer #2: Yes

3. Is the methodology feasible and described in sufficient detail to allow the work to be replicable?

Reviewer #1: Yes

Reviewer #2: Yes

4. Have the authors described where all data underlying the findings will be made available when the study is complete?

Reviewer #1: No

Reviewer #2: Yes

5. Is the manuscript presented in an intelligible fashion and written in standard English?

Reviewer #1: Yes

Reviewer #2: Yes

6. Review Comments to the Author

You may also provide optional suggestions and comments to authors that they might find helpful in planning their study.

Reviewer #1: Thank you for the opportunity to review this interesting manuscript on an important subject. My principal feedback is that there are areas of vague or non-specific language which inhibit the clarity of the paper. At times, especially, opportunities for methodological detail are missed by referring readers to the Micklitz paper or Supplementary material. As a protocol, readers will be interested to read as much detail about the methods as possible. Adding this will enhance the usefulness of the paper. I make some specific suggestions below.

ABSTRACT

This is clear but there are some grammatical errors and missing words. The word "possibly" is rather casual for a protocol and I suggest setting out in a formal manner what the authors propose to do and what caveats may prevent their plans' realisation.

INTRODUCTION

There are some vague expressions: "involved in" IPV, "intertwined with the course of mental distress", "do not use optimal design", "all necessary information", "blurs the evidence", "have effects up to a certain extent". Could the authors be more specific? Intertwined how? Which designs and information are lacking? In what way is the evidence "blurred"? What effects? What extent?

What is meant by ecological bias here?

P4 line 91 "To provide reliable and robust estimates of interventions" - this is a bit vague. Estimates of intervention what - efficacy? In what way will the results be reliable and robust? Perhaps more specific language related to the methods would be more appropriate e.g. more precise? More specific?

Acceptability is grouped together with efficacy but its measurement is likely to be much more heterogeneous than for efficacy; the authors have not commented on this or said what they mean by the term when used quantitatively. Have previous studies been able to apply NMA to an outcome like acceptability?

Likewise, how will safety be conceptualised quantitatively?

P4 RQ2 - while I assume specific mental health conditions will be named in the methods it would help to be more specific about the mental health outcomes of interest. Common mental disorders? Severe mental illness? Any psychiatric diagnosis?

The RQs switch between "efficacy" and "effectiveness"; I suggest consistency throughout.

Given the quantitative methods planned, it seems appropriate to present hypotheses, either in addition or instead of RQs.

METHODS

Line 110 - harms have not previously been mentioned unless this is in reference to "safety". Perhaps the authors can define what they mean by both. Without clarity I would assume that safety refers to whether women report fewer IPV incidents following intervention whereas harms would be adverse effects of the intervention.

Line 115 it would be better to mention the relationship between this and the Micklitz study in the Introduction and explicitly state what that study found and what gaps this study will fill.

Table 1 - it is not clear why psychological and physical risk factors are mentioned in the description of eligible interventions.

The Outcomes box introduces well-being and quality of life outcomes but these constructs are not discussed in the Introduction. Safety and acceptability outcomes are not mentioned in this box but feature in the RQs.

Line 128 - perhaps it would be clearer to say that "their data will be analysed together with" the other studies rather than "they will be merged with"?

Line 133 - given that the purpose of this article is to publish the study protocol I do not think it is enough to refer readers to the Appendix and a different paper for the search strategy. At a minimum I would expect all the search terms to be named.

Line 137 - forward and backward citation tracking?

Line 174 - again, it is not enough to refer readers back to the Micklitz paper. At this point the reader is interested to know which data will be extracted or requested. This is especially important for the RQs on efficacy, effectiveness, acceptability, safety and harms. I note the later section on Outcomes; perhaps these can be combined?

Line 197 - it is not clear why not reporting one characteristic might intrinsically jeopardise the quality of an included study.

Line 215 - "in a sort of "within-study synthesis"" is rather casual phrasing for a protocol. Can the authors describe precisely what they propose to do?

Given that the authors speak about "psychosocial" interventions in the Introduction I wonder whether this term should be defined there, perhaps making reference to the Micklitz taxonomy there instead of late in the Methods.

Line 253 does mindfulness not come under third wave therapies?

Line 269 "which is not ideal" - this is vague. Can the authors specify the problems with heterogeneous outcome measures?

The outcomes section demonstrates that the focus is on common mental disorders, only. I suggest clarifying this throughout the manuscript and justifying at some point why other disorders are not included.

Line 277 - Certain outcomes may be less well known eg space for action. If this is a construct with a validated scale, perhaps the authors can indicate the most widely used in brackets?

Given the range of outcomes to be collected I suggest that the RQs be rewritten to reflect the different sub-groups of outcomes.

Line 306 - "as demonstrated by (42)" is not enough. Please state what will be done.

Line 309 - likewise, "as described in (43)"

Line 316 - "whether... are effect modifiers or prognostic factors": can the authors clarify whether prognostic factors are identified differently to effect modifiers I.e. how this is determined statistically?

Line 322 - again, please elaborate on the Methods rather than referencing study (44).

Line 323 - this is the first time acceptability is defined as the dropout rate. I suggest this needs to be stated in the Introduction to help readers to understand the RQs.

Line 335 - "will be compared" - how? Statistically? In some other way?

I am not a statistics expert but am interested whether corrections are needed for multiple comparisons, based on the number of analyses planned. These may be built into the NMA programme but it would help readers (few of whom will be experts) to be as clear as possible about how the analyses work.

DISCUSSION

Line 341 - I am not sure the planned collaboration with authors needs to be repeated in the Discussion. It would be more helpful to link the planned work to gaps in the literature.

Line 349 - GRADE is presented here as quantifying certainty but in the Methods as determining quality. I suggest consistency of terms throughout.

Line 353 - "some nuancing might be missed" - this is vague. Can the authors specify the potential problems of combining different measures?

Line 356 - my understanding of the Methods was that the Micklitz search would be repeated and not expanded. Can this be clarified and if expanded, please describe how under Methods.

Line 359 - "the field of IPV is highly heterogeneous" - what does this mean? Existing research uses diverse methods and measures? Or IPV affects diverse pateints? Please clarify.

Reviewer #2: l 24 explain this sentence more,as an oversimplification

L28 instead of the word current put a date identifier so future readers do not have to check the publication date to understand the context

Lines 38-43 need expansion - it's not just heterogenous for example but failure to analyse subpopulations, and also for example in some groups the follow up periods are particularly important. The current statements are so generalised as to be unhelpful and almost minority-blind.

L 55 instead of 'was not examined quantitatively' point out that the theoretical approach used - eg realist review - was not appropriate for quantitative evaluation - as it currently reads as a criticism of the previous works rather than a critique

L82 should be expanded on in the limitations in the discussion as also the issue of not getting data - emphasising the considerable efforts being made to do so - yes the authors do include this risk but they need to consider the implications more as this is a major issue with this method. This would also give them the chance to emphasise their good overall design to mitigate and compare.

7. PLOS authors have the option to publish the peer review history of their article (what does this mean? ). If published, this will include your full peer review and any attached files.

**Do you want your identity to be public for this peer review?** For information about this choice, including consent withdrawal, please see our Privacy Policy .

Reviewer #1: **Yes: ** Dr Roxanne Keynejad

Reviewer #2: No

---

## [Author Response · Author response to Decision Letter 1]

10 Dec 2024

Dear Dr Leight and esteemed peer-reviewers,

Firstly, we would like to thank you for the thorough review and insightful comments, we are positive that addressing them will improve the manuscript and our proposed study. In the following, we address the comments one by one:

#Reviewer 1

ABSTRACT

This is clear but there are some grammatical errors and missing words. The word "possibly" is rather casual for a protocol and I suggest setting out in a formal manner what the authors propose to do and what caveats may prevent their plans' realisation.”

We thank the reviewers for this sharp observation. The word “possibly” has been removed, and the planned actions are set out more clearly: “We anticipate that not all study authors will provide access to IPD, which is a limitation of IPDNMA. We aim to address this by combining studies with aggregate data and studies with IPD using Multi-Level Network Meta-Regression (ML-NMR) implemented in multinma. “ We have gone through the manuscript carefully and corrected grammatical errors.

Introduction

“There are some vague expressions: "involved in" IPV, "intertwined with the course of mental distress", "do not use optimal design", "all necessary information", "blurs the evidence", "have effects up to a certain extent". Could the authors be more specific? Intertwined how? Which designs and information are lacking? In what way is the evidence "blurred"? What effects? What extent?”

Thanks for picking this up. We have checked the manuscript carefully and clarified these expressions to be more specific, and have elaborated on all the aforementioned points: “both men and women victims/survivors and perpetrators of IPV”, “The course of IPV affects the course of mental distress and vice versa”, “the most methodologically robust design of a randomised and controlled trial”, “the descriptive statistics that are necessary for a meta-analysis for all the outcomes”, “heterogeneity is an important limitation for these outcomes too”, “. they reduce physical and psychological IPV, but do not reduce sexual IPV and overall IPV”.

“What is meant by ecological bias here?”

We thank the reviewer for giving us the opportunity to highlight a strength of this study. An explanation was added: “Meta-analyses of aggregate data are limited to the outcomes reported in publications and can only explore study-level moderators which often lack statistical power and are vulnerable to ecological bias, where covariate effects at the study-level do not accord with those seen at the individual level”.

“P4 line 91 "To provide reliable and robust estimates of interventions" - this is a bit vague. Estimates of intervention what - efficacy? In what way will the results be reliable and robust? Perhaps more specific language related to the methods would be more appropriate e.g. more precise? More specific?”

Thank you for pointing this out. We have re-worded the statement of the aims to be clearer: ”This study aims to provide more valid and detailed estimates of the clinically relevant benefits and harms than the existing pairwise meta-analyses, as well as moderators to that, of each type of psychosocial intervention comparatively by obtaining the IPD of the studies to enable exploration and adjustment of effect modifiers. We will fulfil this aim by conducting an IPDNMA, to estimate the relative effectiveness of multiple psychosocial interventions for mental health, psychosocial well-being, IPV, safety behaviours, intervention dropout and harms (i.e. adverse effects of the interventions. We also aim to estimate study-level and individual-level moderator effects to identify which psychosocial intervention suits individual IPV survivors best.”

“Acceptability is grouped together with efficacy but its measurement is likely to be much more heterogeneous than for efficacy; the authors have not commented on this or said what they mean by the term when used quantitatively. Have previous studies been able to apply NMA to an outcome like acceptability?

Likewise, how will safety be conceptualised quantitatively?”

We thank the reviewer for pointing this out. Yes, previous NMAs of psychotherapy have had acceptability as an outcome, but they operationalised it as all-cause dropout, due to heterogeneity in reporting (Cuijpers et al, 2019; Cuijpers et al, 2021). We hope that with the IPD we will be able to overcome that for at least some studies. We will use intervention dropout as the closest available quantitative indication of acceptability, as opposed to study dropout, because we believe that dropping out from research assessments is different from skipping intervention sessions, and the former has hardly, if any, clinical relevance. Given that intervention dropout is merely an indication of acceptability and it might not accurately reflect it, we now refer to intervention dropout as our study outcome instead of intervention acceptability.

We now refer to reduction in IPV and increase in safety behaviours as our study outcomes, instead of “safety” only, to avoid confusion:

“Which victim/survivor and intervention characteristics moderate the comparative effectiveness of each set of intervention components that have been tested together in reducing IPV and improving safety behaviours of IPV victims/survivors?”

“Which victim/survivor and intervention characteristics moderate the dropout from each intervention?

“P4 RQ2 - while I assume specific mental health conditions will be named in the methods it would help to be more specific about the mental health outcomes of interest. Common mental disorders? Severe mental illness? Any psychiatric diagnosis?”

We thank the reviewer for this sharp observation. We are interested in all mental health-related outcomes, but most often this is symptoms of depression and/or anxiety, post-traumatic stress, and substance use, therefore we set these as our primary outcomes. We anticipate that there will be too scarce evidence available for other outcomes to support any analysis. We now specify that from the research question already: “Which victim/survivor and intervention characteristics moderate the comparative effectiveness of each set of intervention components that have been tested together in improving commonly tested mental health and psychosocial well-being outcomes of IPV victims/survivors?”

“The RQs switch between "efficacy" and "effectiveness"; I suggest consistency throughout.”

We thank the reviewer for this comment. We have now consistently used “effectiveness” throughout the manuscript. We will include all RCTs comparing psychosocial interventions with a standard care control condition or another psychosocial intervention, and the majority of included studies will be effectiveness studies, so this is the most appropriate term.

“Given the quantitative methods planned, it seems appropriate to present hypotheses, either in addition or instead of RQs.”

We understand this comment and we thank the reviewer, however meta-analyses are retrospective explorative investigations. Hypothesis are more common in prospective experimental designs. We have not made any changes in the text based on this comment.

METHODS

“Line 110 - harms have not previously been mentioned unless this is in reference to "safety". Perhaps the authors can define what they mean by both. Without clarity I would assume that safety refers to whether women report fewer IPV incidents following intervention whereas harms would be adverse effects of the intervention.”

We thank the reviewer for giving us the opportunity to make this important distinction. The harms are not related to safety, they are adverse intervention effects. They are now mentioned and referred to as “adverse effects if the interventions”.

“Line 115 it would be better to mention the relationship between this and the Micklitz study in the Introduction and explicitly state what that study found and what gaps this study will fill.”

We thank the reviewer for this suggestion, we have now added a paragraph in the introduction, right before the aims: “IPDNMA has the potential to address many of the limitations and issues raised by one of the most recent meta-analyses with a very close scope to this one (16). Micklitz and colleagues found that psychosocial interventions, and especially interventions combining advocacy with psychotherapy, i.e. integrative interventions, had a small yet statistically significant effect in reducing depression and post-traumatic stress symptoms, compared to waiting list or inactive control conditions. However, there was considerable heterogeneity even in the subgroup analyses, and half the studies could not be included in the quantitative meta-analysis because the necessary statistics were not reported at the publications of the primary studies (16). In addition, the authors highlighted the need to test participant-level moderators of effect (16). We aim to build on the comprehensive scope and rigour of the study by Micklitz and colleagues by updating and adapting their searches, and seeking IPD from the included studies. Given that most RCTs have used Treatment As Usual (TAU) control conditions, or participants were most commonly allowed to seek other help, and also that we are going to compare interventions, we are going to estimate the effectiveness of the interventions, i.e. compared to a real-world standard, rather than efficacy, i.e. that the interventions have some effect compared to nothing, which is usually a sterilised control condition.”

“Table 1 - it is not clear why psychological and physical risk factors are mentioned in the description of eligible interventions.

The Outcomes box introduces well-being and quality of life outcomes but these constructs are not discussed in the Introduction. Safety and acceptability outcomes are not mentioned in this box but feature in the RQs.”

We thank the reviewer for this suggestion. The text on psychological and physical risk factors has been removed to avoid confusion, and explanations have been added to the “Participants” and “Outcomes” sections of the table:

Table 1. Eligibility criteria by PICO

Participants Individuals or couples reporting either past or ongoing IPV at randomisation, but studies solely on those involved in perpetration are excluded. Studies are deemed partially eligible if they did not recruit participants based on IPV status, but they measured IPV at baseline, and a proportion of the participants had been victimised at baseline. The IPD only of the victimised participants will be requested. If it is not possible to obtain the IPD of a partially eligible study, and outcome measures separately for victimised participants are not reported in the publication, the study will not be included in the analysis at all.

Interventions Psychosocial interventions aimed at IPV survivors using any mode of delivery at any setting. Both psychotherapeutic and advocacy interventions are deemed as psychosocial. Interventions aimed solely at perpetrators, primary prevention to individuals who have not experienced IPV at baseline, and physical safety (such as being in a refuge/shelter and police interventions) are not included.

Comparators All comparators are eligible, including no intervention, waiting list, Treatment as Usual (TAU), another active intervention, placebo or sham version of the intervention. The interventions will be categorised and classified into nodes of the network based on their description regardless of whether they were tested as control or active condition in the original studies.

Outcomes Eligible outcomes are all outcomes related to mental health (e.g. depression) as well as psychosocial outcomes (e.g. empowerment and social support) . Studies measuring only IPV are not eligible, but if the included studies have measured IPV, it will be analysed. Likewise, studies are not included based on whether they measured intervention dropout, and health service use, but these outcomes will be analysed.

“Line 128 - perhaps it would be clearer to say that "their data will be analysed together with" the other studies rather than "they will be merged with"?”

We thank the reviewer for this suggestion. The text was rephrased accordingly: “Only randomised controlled trials (RCTs) are eligible for inclusion, including pilot RCTs and studies with few participants, as in most cases they will be analysed together with larger studies testing a similar intervention.”

“Line 133 - given that the purpose of this article is to publish the study protocol I do not think it is enough to refer readers to the Appendix and a different paper for the search strategy. At a minimum I would expect all the search terms to be named.”

We thank the reviewer for this important addition. The full search string was added to the manuscript as a table.

intimate partner violence OR spouse abuse OR domestic violence OR gender-based violence OR rape OR battered women OR "intimate partner violence" OR "partner violence" OR "interpersonal violence" OR "dating violence" OR "domestic violence" OR "family violence" OR "gender-based violence" OR "gender based violence" OR "violence in close relationships" OR "intimate partner abuse" OR spous* abuse OR "abusive relationship" OR "marital abuse" OR "marital rape" OR "dating abuse" OR "intimate partner aggression" OR "battered women" OR "battered wives" AND Mental health services OR Psychotherapy OR Couples Therapy OR Counselling OR Psychosocial intervention OR Social Work, Psychiatric OR Psychiatric Rehabilitation OR Therapy, computer-assisted OR Telemedicine OR Internet-Based Intervention OR Distance counselling OR "intervention" OR "treatment" OR "program" OR "therap*" OR "psychotherap*" OR "counsel*" OR "support service*" OR "advocacy" OR "secondary prevention" OR "tertiary prevention" OR "rehabilitation" AND randomized controlled trials as topic OR clinical trials as topic OR "randomized controlled trial" OR "clinical trial" OR "controlled clinical trial" OR "clinical trial" OR "clinical trial protocol" OR "clinical study" OR RCT OR random* OR trial

“Line 137 - forward and backward citation tracking?”

We thank the reviewer for this opportunity to clarify. An explanation was added to the manuscript: “the reference lists of the included studies will be checked for eligible papers, as well as which studies cite the eligible studies”.

“Line 174 - again, it is not enough to refer readers back to the Micklitz paper. At this point the reader is interested to know which data will be extracted or requested. This is especially important for the RQs on efficacy, effectiveness, acceptability, safety and harms. I note the later section on Outcomes; perhaps these can be combined?”

We thank the reviewer for this comment. The data that are extracted for this study are listed in the paragraph “Data extraction of study characteristics and aggregate data: “The study characteristics to be extracted in a spreadsheet are: author, publication year, citation, country, recruitment setting, trial setting, number of trial arms, number of follow-ups, timing of follow-ups, timing of post-treatment, primary and secondary outcomes, type of randomisation, unit of allocation, method of randomisation, sample size in total and in each arm, treatment and study dropout, intention-to-treat or per-protocol analysis, age, gender, current frequency and/or severity of IPV, past frequency and/or severity of IPV, specific target population, eligibility criteria, description of intervention, theoretical orientation, delivery mode, type of therapists/facilitators, number and duration of sessions, frequency of sessions, description of control, whether adverse events and adherence to the intervention were reported, conclusions, the means and standard deviations of all relevant outcomes at all timepoints, or which statistics were reported instead of mean and standard deviation.”

We do not refer to the Micklitz paper to report which data are extracted, we are simply explaining that we will make use of the data Micklitz and colleagues have already extracted for the studies they included, instead of extracting them again. The text had been rephrased to clarify: “The characteristics and aggregate data that have already been extracted by Micklitz et al (16) for the

---

## [Decision Letter · Decision Letter 1]

14 Jan 2025

PONE-D-24-25327R1Individual Participant Data Network Meta-analysis of psychosocial interventions for survivors of intimate partner violence: Study protocolPLOS ONE

Dear Dr. Palantza,

Thank you for submitting your manuscript to PLOS ONE.  I believe it is nearly ready for publication, but one referee has flagged some further very minor points related to language and typographical errors.  Kindly address these minor errors and resubmit, and the manuscript can then proceed to the acceptance phase.

We look forward to receiving your revised manuscript.

Kind regards,

Jessica Leight, PhD

Academic Editor

PLOS ONE

**Journal Requirements:**

Reviewers' comments:

Reviewer's Responses to Questions

**Comments to the Author**

1. Does the manuscript provide a valid rationale for the proposed study, with clearly identified and justified research questions?

Reviewer #1: Yes

Reviewer #2: Yes

2. Is the protocol technically sound and planned in a manner that will lead to a meaningful outcome and allow testing the stated hypotheses?

Reviewer #1: Yes

Reviewer #2: Yes

3. Is the methodology feasible and described in sufficient detail to allow the work to be replicable?

Reviewer #1: Yes

Reviewer #2: Yes

4. Have the authors described where all data underlying the findings will be made available when the study is complete?

Reviewer #1: Yes

Reviewer #2: Yes

5. Is the manuscript presented in an intelligible fashion and written in standard English?

Reviewer #1: Yes

Reviewer #2: Yes

6. Review Comments to the Author

You may also provide optional suggestions and comments to authors that they might find helpful in planning their study.

**Reviewer #1: ** Thank you to the authors for their clear and comprehensive response to my feedback, which I believe has enhanced the clarity of the paper.

The authors have focused on effectiveness rather than efficacy. I just wanted to confirm/clarify this choice, given that efficacy is usually reported by studies under idealised, research study conditions and (given the inclusion criteria) I would have thought that they will identify evidence like this rather than real-world effectiveness data.

A few small points:

References to the Micklitz review have been moved up in the Introduction but as a result the first reference to that paper sounds like readers are assumed to be familiar with it. The phrasing could be tweaked so that the first time the Micklitz paper is mentioned, it summarises what that paper is and how it relates to this study.

Although the title has changed to individual participant instead of patient, the term patient has persisted, e.g. line 140 - this should be harmonised across the manuscript.

Line 162-165: this sentence is long and the meaning of the phrase "moderators to that" was not clear to me.

There are a few errors arising from the corrections e.g. line 166 "we will fulfil this aim by to conducting...", line 377 "componentss", "studies with aggregate data studies"

The tense changes at times between present and future, e.g. Table 1 says both "studies are deemed partially eligible" but also "will not be included".

I believe the term used by GRADE is certainty "of" the evidence rather than "in".

Line 384-388 - this is a long sentence with several clauses that could be broken into two to aid clarity.

**Reviewer #2: ** Thank you for responding to my queries to my overall satisfaction. While it is good to be flexible and improve the study design, any changes once the study has begun/been funded should be listed under ‘changes to protocol’ to be compared with the original, as a key reason for publishing protocols, for audit. The editor may wish to follow this point up. Good luck with the analysis and data collection.

7. PLOS authors have the option to publish the peer review history of their article (what does this mean? ). If published, this will include your full peer review and any attached files.

**Do you want your identity to be public for this peer review?** For information about this choice, including consent withdrawal, please see our Privacy Policy .

Reviewer #1: **Yes: ** Dr Roxanne Keynejad

Reviewer #2: No

---

## [Author Response · Author response to Decision Letter 2]

7 Feb 2025

Dear Dr Leight and dear reviewers,

Thank you very much for thoroughly reviewing the updated version of our manuscript and for picking up on details to ensure that the manuscript is flawless. Below is a point by point reply to all the comments:

Reviewer 1

“The authors have focused on effectiveness rather than efficacy. I just wanted to confirm/clarify this choice, given that efficacy is usually reported by studies under idealised, research study conditions and (given the inclusion criteria) I would have thought that they will identify evidence like this rather than real-world effectiveness data.”

We thank the reviewer for the opportunity to better justify our choice to use the term effectiveness instead of efficacy. Based on the criteria described by Gartlehner et al in 2006, we are confident that our study measures the effectiveness rather than the efficacy, because:

a) the population is usually recruited from general healthcare or even community settings

b) the eligibility criteria are typically not very refined

c) the outcome criterion varies a lot across trials as subjective, objective and health outcomes have been measured by many of the trials.

d) The study lengths are typically closer to the length of treatment in a clinical setting, and compliance/adherence to the intervention is an outcome measure in our study

e) Adverse events have been reported by less than half of the trials, so this criterion is unclear

f) Sample sizes also differ greatly, so this criterion is unclear too

g) Many of the studies have followed the intention-to-treat principle

In addition, the eligibility criteria of our study allow the inclusion of any control condition, and most of these control conditions are standard/routine care, also commonly referred to as “treatment-as-usual”.

“References to the Micklitz review have been moved up in the Introduction but as a result the first reference to that paper sounds like readers are assumed to be familiar with it. The phrasing could be tweaked so that the first time the Micklitz paper is mentioned, it summarises what that paper is and how it relates to this study.”

We thank the reviewer for pointing this out, we have now rephrased the sentence to make sure the Mickplitz paper is introduced properly:

“Micklitz and colleagues in their meta-analysis, which is one of the most recent and comprehensive meta-analyses on IPV and mental health, categorised the tested psychosocial interventions into three main categories: psychological, advocacy-oriented and integrated (a combination of the two) (16).”

“Although the title has changed to individual participant instead of patient, the term patient has persisted, e.g. line 140 - this should be harmonised across the manuscript.”

We thank the reviewer for picking up this important detail. We have now replaced the word “patient” with that of “participant”, except for the term “Patient and public involvement”, because this is a standard term.

“There are a few errors arising from the corrections e.g. line 166 "we will fulfil this aim by to conducting...", line 377 "componentss", "studies with aggregate data studies"”

We thank the reviewer for spotting these mistakes. We have now corrected them.

“The tense changes at times between present and future, e.g. Table 1 says both "studies are deemed partially eligible" but also "will not be included".”

We thank the reviewer for stressing this issue, we have now made sure the tense use is consistent.

“I believe the term used by GRADE is certainty "of" the evidence rather than "in".”

We thank the reviewer for picking up another important detail. We have corrected it throughout the manuscript.

“Line 162-165: this sentence is long and the meaning of the phrase "moderators to that" was not clear to me.”

We thank the reviewer for raising this point. We have now rephrased this sentence to:

“This study aims to provide more valid and detailed estimates of the comparative clinically relevant benefits and negative effects of each type of psychosocial interventions than the existing pairwise meta-analyses. The IPD of the studies will be obtained to enable this estimation, as well as exploration and adjustment of effect modifiers.”

“Line 384-388 - this is a long sentence with several clauses that could be broken into two to aid clarity.”

We thank the reviewer for giving us the opportunity to clarify even further. We have rephrased the sentence to:

“We will also create pseudo-IPD by simulating from a model using the reported baseline summaries and regression coefficients for effect modifiers as described in (47), if available in the publications of studies that do not provide IPD.”

Reviewer 2

“Thank you for responding to my queries to my overall satisfaction. While it is good to be flexible and improve the study design, any changes once the study has begun/been funded should be listed under ‘changes to protocol’ to be compared with the original, as a key reason for publishing protocols, for audit. The editor may wish to follow this point up. Good luck with the analysis and data collection.”

We thank the reviewer for reminding us and the journal of this crucial aspect. We now include a section listing these changes and we mention the shift in the data reduction method and the increased focus on the comparison of components:

“Changes to the protocol: Since the awarding of the funding and the beginning of the study, a few changes have been made to the protocol. The data reduction method has changed from a lasso method to assessment of collinearity, and more emphasis has been placed on the comparison of intervention components, as it appears to be important for the overall objective of disentangling effectiveness.”

We are confident that these revisions render the protocol ready for publication.

Kind regards,

Christina Palantza

---

## [Editor Report · Decision Letter 2]

11 Feb 2025

Individual Participant Data Network Meta-analysis of psychosocial interventions for survivors of intimate partner violence: Study protocol

PONE-D-24-25327R2

Dear Dr. Palantza,

We’re pleased to inform you that your manuscript has been judged scientifically suitable for publication and will be formally accepted for publication once it meets all outstanding technical requirements.

Kind regards,

Jessica Leight, PhD

Academic Editor

PLOS ONE
---

## [Editor Report · Acceptance letter]

PONE-D-24-25327R2

PLOS ONE

Dear Dr. Palantza,

I'm pleased to inform you that your manuscript has been deemed suitable for publication in PLOS ONE. Congratulations! Your manuscript is now being handed over to our production team.

Kind regards,

on behalf of

Dr. Jessica Leight

Academic Editor

PLOS ONE